# Single-cell co-mapping reveals relationship between chromatin state and gene expression in early zebrafish development

Vivek Bhardwaj[1,2*†‡], Alberto Griffa[1,2†], Helena Viñas Gaza[1,2], Peter Zeller[1,2], Alexander van Oudenaarden[1,2*]

[1]Hubrecht Institute-KNAW (Royal Netherlands Academy of Arts and Sciences), Oncode Institute, Utrecht, Netherlands; [2]University Medical Center Utrecht, Utrecht, Netherlands

*For correspondence:
v.bhardwaj@uu.nl (VB);
a.vanoudenaarden@hubrecht.
eu (AO)

†These authors contributed
equally to this work

Present address: ‡Institute of
Biodynamics and Biocomplexity,
Department of Biology, Utrecht
University, Utrecht, Netherlands

Competing interest: The authors
declare that no competing
interests exist.

Reviewing Editor: H Efsun Arda,
National Cancer Institute, United
States

## eLife Assessment

In this **valuable** study, the authors examine transcription and chromatin dynamics during early zebrafish development by simultaneously profiling histone modifications and full-length transcriptomes in thousands of single cells, providing **solid** analysis that chromatin and transcriptional states are initially weakly correlated in early embryonic cells and become progressively more aligned as differentiation proceeds. The work also supports a model in which promoter-anchored cis-spreading of H3K27me3 contributes to stable gene silencing during development. Future functional perturbations and orthogonal validations will be needed to determine the causal contribution of Polycomb spreading to fate commitment. Overall, the dataset and accompanying analyses provide a robust resource and a quantitative framework for studying chromatin-transcription relationships during vertebrate embryogenesis.
[Editors note: this paper was reviewed by Review Commons.]

**Abstract** Establishing a cell type-specific chromatin landscape is crucial for the maintenance of cell identity during embryonic development. However, our knowledge of how this landscape is set during vertebrate embryogenesis has been limited, due to the lack of methods to jointly detect chromatin modifications and gene expression in the same cell. Here we present a multimodal measurement of full-length transcriptome and histone modifications in individual cells during early embryonic development in zebrafish. We show that before the formation of germ layers, the chromatin and transcription states of cells are uncoupled and become progressively connected during gastrulation and somitogenesis. Silencing of developmental genes is achieved by local spreading of repressive chromatin together with cell type-specific demethylation. Combining transcription factor (TF) expression and chromatin states within an interpretable machine learning model, we classify TFs as lineage-specific activators and repressors and identify a subset of TFs that are epigenetically regulated. Altogether, our data resolves the dynamic relationship between chromatin and transcription during early vertebrate development and clarifies how these two layers interact to establish cell identity.

## Introduction

Early embryonic development in animals is characterized by the controlled movement and positioning of cells, establishment of a body plan, and specification of tissue-specific cell states. While the spatial gradients of morphogens dominate the former two events (*Xu et al., 2014*), the maintenance of cell

identity is believed to be mainly regulated by chromatin state (*Bogdanović et al., 2012*). Similar to the morphogens that regulate the patterning of an embryo, the chromatin state can also be transgenerationally inherited (*Fitz-James and Cavalli, 2022*). This might play an important role in predefining the spatiotemporal expression of genes during early development and regulate cell fates. The relationship between chromatin state and gene expression has been studied using whole-genome assays applied to cultured cell populations, whole tissues, or enriched cell populations sorted using cell surface or transgenic markers (*Abascal et al., 2020*; *Kundaje et al., 2015*). Recently, chromatin and DNA methylome mapping techniques have been developed to resolve cellular heterogeneity of epigenetic states at the level of individual cells. Major progress has been made using DNA methylome profiling of single cells, which have mostly been applied to study adult tissues (*Bai et al., 2025*; *Liu et al., 2021*; *Nichols et al., 2022*). We and others have applied single-cell methods to study chromatin states in adult tissues (*Bartosovic et al., 2021*; *Cheung et al., 2018*; *Wu et al., 2021*; *Zeller et al., 2023*). However, genome-wide studies mapping temporal chromatin changes of single cells during early embryogenesis are rare (*Argelaguet et al., 2019*; *Clark et al., 2022*; *Fu et al., 2025*; *Guo et al., 2017*; *Liu et al., 2025*; *Zhao et al., 2022*). This leaves a gap in our understanding of the process of establishment and propagation of cell type-specific chromatin states during early embryonic development.

In this study, we asked how the active and silenced chromatin states of cells are shaped during early vertebrate embryogenesis, using zebrafish as a model system. As the chromatin and gene expression are highly dynamic in embryos across cells, bulk assays would average out these biologically important differences. Similarly, a single-cell assay profiling either chromatin or transcriptome alone cannot measure how these two layers interact with each other during development in a single cell. Therefore, we applied our single-cell co-mapping assay T-ChIC (*Zeller et al., 2024*) to jointly profile the genome-wide active and silencing histone modifications together with full-length transcriptome from the same single cells during early zebrafish development (4–24 hpf). Using this data, we infer continuous developmental trajectories and ask how the chromatin state correlates with the expression of transcription factors and other developmentally important genes during cell fate commitment.

## Results
### Paired profiling of histone modifications and transcriptome of single cells during zebrafish embryogenesis

We recently developed a single-cell multi-omics method, termed T-ChIC (transcriptome and chromatin immuno-cleavage), which extends the previously described sortChIC (*Zeller et al., 2023*) and VASA-seq (*Salmen et al., 2022*) protocols, by integrating them in a single workflow (*Zeller et al., 2024*). This allows us to quantify the pattern of histone modifications at kilobase resolution, while simultaneously providing full-length transcriptome coverage in single cells. To apply T-ChIC to study multiple time points across early zebrafish development, we extended this protocol with an optimized cell dissociation and sample multiplexing strategy that allows collection of embryos from different time points of an experiment while reducing batch effects (*Figure 1a*, 'Materials and methods'). We applied this modified workflow, termed 'whole-organism T-ChIC' (woT-ChIC), to quantify the polycomb complex-mediated histone mark, H3K27me3, in zebrafish embryos collected at six selected time points post-fertilization, obtaining a total of 18,432 cells. This dataset provides us with complete coverage of gastrulation (4, 6, 8, 10 hpf), along with the beginning and the end of somitogenesis (12 and 24 hpf, respectively, *Figure 1b*). We produced the woT-ChIC dataset in four independent biological replicates, along with two additional replicates that contained cells without a functional antibody, to validate data quality (*Supplementary file 1*). This subset (labeled 'T-noChIC') showed a similar number of detected transcripts and was co-clustered with woT-ChIC cells, confirming that the transcriptome quality of woT-ChIC is independent of the ChIC fraction (*Figure 1—figure supplement 1a and b*). After removing cells with low numbers of MNase cuts and potentially over-fragmented cells, we observed a strong enrichment of chromatin signal over specific genomic regions (*Figure 1—figure supplement 1d and e*).

Early zebrafish embryos contain a high load of maternal transcripts required for early embryonic development, which are temporally replaced with newly transcribed, zygotic RNA (*Fishman et al., 2023*). Consistent with this transition, we observed a substantial decrease in unique fragment counts from spliced reads compared to unspliced reads with developmental time (*Figure 1—figure*

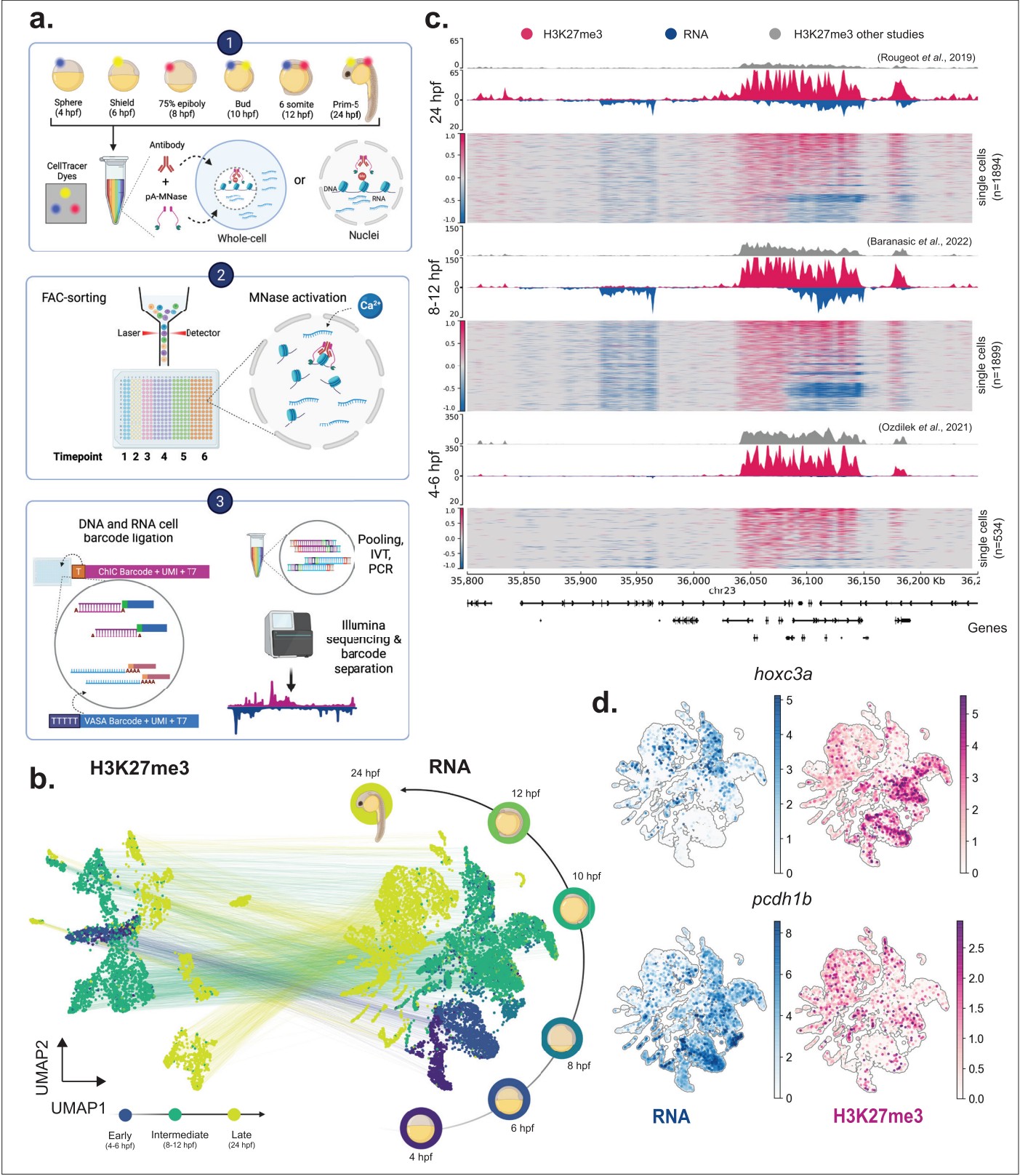

**Figure 1.** Whole-organism T-ChIC of zebrafish embryos quantifies full-length transcripts and histone modifications from the same single cell. (**a**) woT-ChIC experimental workflow: (1) single cells from different time points are labeled with a combination of CellTracer dyes and permeabilized either to retain cytoplasmic RNA (whole-cell), or exclude it (nuclei) before antibody + pA-MNase incubation. (2) Cells are pooled and sorted in 384-well plates with position-indexed RNA and ChIC barcodes. Addition of Ca^++ activates the MNase to cut on the target regions. (3) DNA and RNA fragments are

Figure 1 continued

ligated and repaired before re-pooling them for IVT and PCR amplification to produce sequencing libraries. (**b**) UMAP projections of single cells using signals from H3K27me3 (left) and transcriptome (right) and colored by timepoints. The six sampled time points (right) are pooled into three groups (early/middle/late) based on the complexity of H3K27me3 signal. (**c**) Single-cell track plot showing signal on the 450 kb region around the *hoxc* gene cluster. The heatmaps show signals (read counts) in single cells (capped from -1 to +1, where -ve signal shows RNA counts and +ve signal shows ChIC counts). The coverage tracks on top show the pseudo-bulk signal (blue: RNA, pink: H3K27me3). Publicly available bulk H3K27me3 datasets are shown for comparison (gray). (**d**) UMAP projections (based on transcriptome signal) showing gene-level normalized signal for RNA (blue) or H3K27me3 (pink) on two selected genes.

The online version of this article includes the following figure supplement(s) for figure 1:

**Figure supplement 1.** Quality control for the RNA and ChIC fraction.

**Figure supplement 2.** Evaluation of the quantitative chromatin signal from woT-ChIC.

*supplement 1f*). Despite these dynamics, our overall number of detected genes with both spliced and unspliced counts is higher than previously reported in scRNA-seq studies (*Farrell et al., 2018*; *Wagner et al., 2018*) due to the increased sensitivity and full-length RNA recovery (*Supplementary file 2*). Moreover, with our total RNA profiling approach, we were also able to detect developmentally important non-coding RNAs such as miR-430 (in pluripotent cells) known to be critical for clearance of maternal RNA in zebrafish (*Giraldez et al., 2006*; *Liu et al., 2020*), and miR-124 (in neural ectoderm), a known regulator of neuronal differentiation which is conserved across species (*Gourishetti et al., 2023*; *Figure 1—figure supplement 1c*). At the chromatin level, we observed increased MNase cuts (representing H3K27me3 signal) per cell as development progresses (*Figure 1—figure supplement 1g*). This corroborates previous observations of increasing H3K27me3 abundance during development based on bulk chromatin assays (*de la Calle Mustienes et al., 2015*; *Vastenhouw et al., 2010*). Overall, we retained 9275 cells with both transcriptome and H3K27me3 signal for further analysis.

We divided our data into early (4–6 hpf), intermediate (8, 10, 12 hpf), and late (24 hpf) time points, and compared our H3K27me3 signal with publicly available datasets at the corresponding time points (*Figure 1c*, 'Materials and methods'). While our pseudo-bulk H3K27me3 profiles showed a high genome-wide correlation with publicly available bulk ChIP data from matched time points (*Figure 1—figure supplement 2a*), the analysis of genomic bins ranked by H3K27me3 signal shows improved signal enrichment of our data relative to the publicly available bulk sequencing data at a comparable sequencing depth (*Figure 1—figure supplement 2b*). Moreover, H3K27me3 levels show a clear relationship with the silencing of associated genes in single cells at all time points (*Figure 1d*, *Figure 1—figure supplement 2c*). We observed high H3K27me3 levels associated with the silencing of gene expression as early as 4 hpf for *hoxc3a*, involved in anterior-posterior patterning, as well as silencing of genes such as *pcdh1b* late during development (*Figure 1d*, *Figure 1—figure supplement 2c*). Furthermore, we also observed a transient association of H3K27me3 on genes. For example, *rfx4*, expressed in the central nervous system and neural rod, was silenced in non-neural ectoderm cells by H3K27me3 during gastrulation (*Figure 1—figure supplement 2c and d*). These results suggest that our data allows us to gain quantitative insight into the relationship between H3K27me3 and gene expression during development.

## Spatiotemporal spreading of H3K27me3 associates with the silencing of gene expression during development

To annotate cell types in our data, we performed Leiden clustering of cells using their gene expression signal, followed by canonical correlation analysis of gene expression with that of a previously published time-course scRNA-seq data set (*Wagner et al., 2018*; *Figure 2—figure supplement 1a*, 'Materials and methods'). Virtually all our cells matched one of the annotated cells from Wagner et al. with high confidence, allowing successful label transfer into our data (*Figure 2—figure supplement 1b and c*). We further refined these labels based on cell ontologies from the Zebrafish Information Network (*Bradford et al., 2022*), to categorize our cells into 34 cell types (*Figure 2—figure supplement 1e*). Cell type proportions were consistent between the biological replicates of woT-ChIC (*Figure 2—figure supplement 1f*). To get a fine-grained view of cellular heterogeneity while reducing signal dropouts, we aggregated cells that are transcriptionally similar to each other into 'metacells' (*Persad et al., 2023*; *Figure 2—figure supplement 1d*, 'Materials and methods'). Interestingly, most of the cell types annotated based on their gene expression profiles also show a clear separation

based on their H3K27me3 enrichment as early as 8–12 hpf, suggesting that distinct, cell-state-specific H3K27me3 patterns already start to appear during gastrulation (*Figure 2a*).

Next, we asked how the global H3K27me3 landscape is established in the cells during lineage commitment. We observed that with time, the number of genomic bins with H3K27me3 signal increased. In contrast, the average signal in detected bins plateaued at 24 hpf, indicating new regions acquiring H3K27me3 signal instead of enrichment of signal on pre-marked regions (*Figure 2—figure supplement 2a*). Therefore, we asked whether this increase in signal comes from a de novo gain in H3K27me3 or as a result of spill-over (indicating 'cis-spreading') of increasing H3K27me3 density from previously enriched regions. At least a subset of enriched regions in pluripotent cells displays a cis-spreading of signal with differentiation, covering developmentally important genes such as the *zic* locus (*Figure 2b*). To quantify cis-spreading genome-wide, we first subsetted the genomic bins, which had signal in at least 5% of filtered *Pluripotent* cells (at 4 hpf). Apart from tightly repressed genes such as *gata3, nr2f1a, six3b, pax9a, foxc1b, zic1/4, hox* clusters, and the *pcdh1/2* cluster with a broadly distributed signal, all other H3K27me3 signal was localized within 5 kb bins and the majority (65%) of these bins overlapped with a promoter region. We then calculated the signal on these bins compared to the background signal (averaged over 100 kb region) surrounding these bins in single cells ('Materials and methods'). These two signals correlated positively for about 30% of the 5 kb bins, suggesting a spillover from the main signal peak to the surrounding background. In contrast, the remaining 70% displays a low correlation with background indicating a localized enrichment without spill-over to the surrounding background (*Figure 2c*, *Figure 2—figure supplement 2b*).

We confirmed this enrichment with an alternative approach based on domain calling on the pooled, pseudo-bulk dataset ('Materials and methods'). While wider H3K27me3 domains detected on the pooled data correspond to the signal at 24 hpf, the sharper subpeaks within those domains were observed at early (4–6 hpf) time points (*Figure 2—figure supplement 2d*). Relatively mature cell types, particularly from the neural ectoderm (such as differentiating neurons), show a higher correlation of subpeaks to the background, suggesting a wider spread in signal (*Figure 2—figure supplement 2e*). We further stratified this signal in search of bins with a significant difference between lineages ('Materials and methods'). We only detected a handful of bins with statistically significant differences between lineages, and the mean H3K27me3 signal indicated that these results are not robust (*Figure 2—figure supplement 2c*). Therefore, the spread of H3K27me3 signal does not appear to be lineage-specific. To test whether this characteristic might be conserved across species, we re-analyzed a previously published ChIP-seq dataset of mouse embryos (*Xiang et al., 2020*; 'Materials and methods'). Comparing H3K27me3 signals at and around gene promoters between E5.5 epiblast and post-gastrulation ectoderm lineage, we see the evidence of signal spreading from a subset of these sites (*Figure 2—figure supplement 2f*), suggesting that the spread of H3K27me3 signal from promoters might be a conserved phenomenon.

To understand how this spread of H3K27me3 relates to gene expression in time, we plotted the expression of the 'host' gene (genes with promoter enrichment of H3K27me3 in pluripotent cells), and the 'nearby' genes (with promoter within 100 kb region) over single cells arranged in pseudotime (*Figure 2d*). Interestingly, the 'host' genes displayed an increased expression before the spread of H3K27me3 signal, followed by silencing post-spreading (*Figure 2d*). In contrast, the 'nearby' genes displayed relatively smaller changes in transcription during this process but were also downregulated at a later stage (*Figure 2—figure supplement 2g*). To identify genes whose expression is silenced as a result of spreading of H3K27me3, we applied linear regression to predict gene expression as a function of H3K27me3 density (defined as the number of reads per kb) on their nearest, or overlapping domains in metacells ('Materials and methods'). Silenced genes showed a strong negative correlation of H3K27me3 density with their expression, with the strongest targets being *hox* and *pcdh1* gene clusters (*Figure 2—figure supplement 2h*).

Considering most genes seem to be in the process of gaining H3K27me3 until 24 hpf, we asked whether there is a subset of genes that lose H3K27me3. For this, we performed a differential H3K27me3 signal analysis for each cell type at 24 hpf stage compared to cell types from earlier stages and selected genes with a significant loss of H3K27me3 in specific cell types (log2-FC < –1, FDR < 0.05). We identified 265 genes across 10 cell types, with most genes being detected in periderm and differentiating neurons (*Figure 2e*, *Supplementary file 3*). For almost all of these genes, we observed that the H3K27me3 was specifically lost in their cell type of origin, while being gained in almost all other

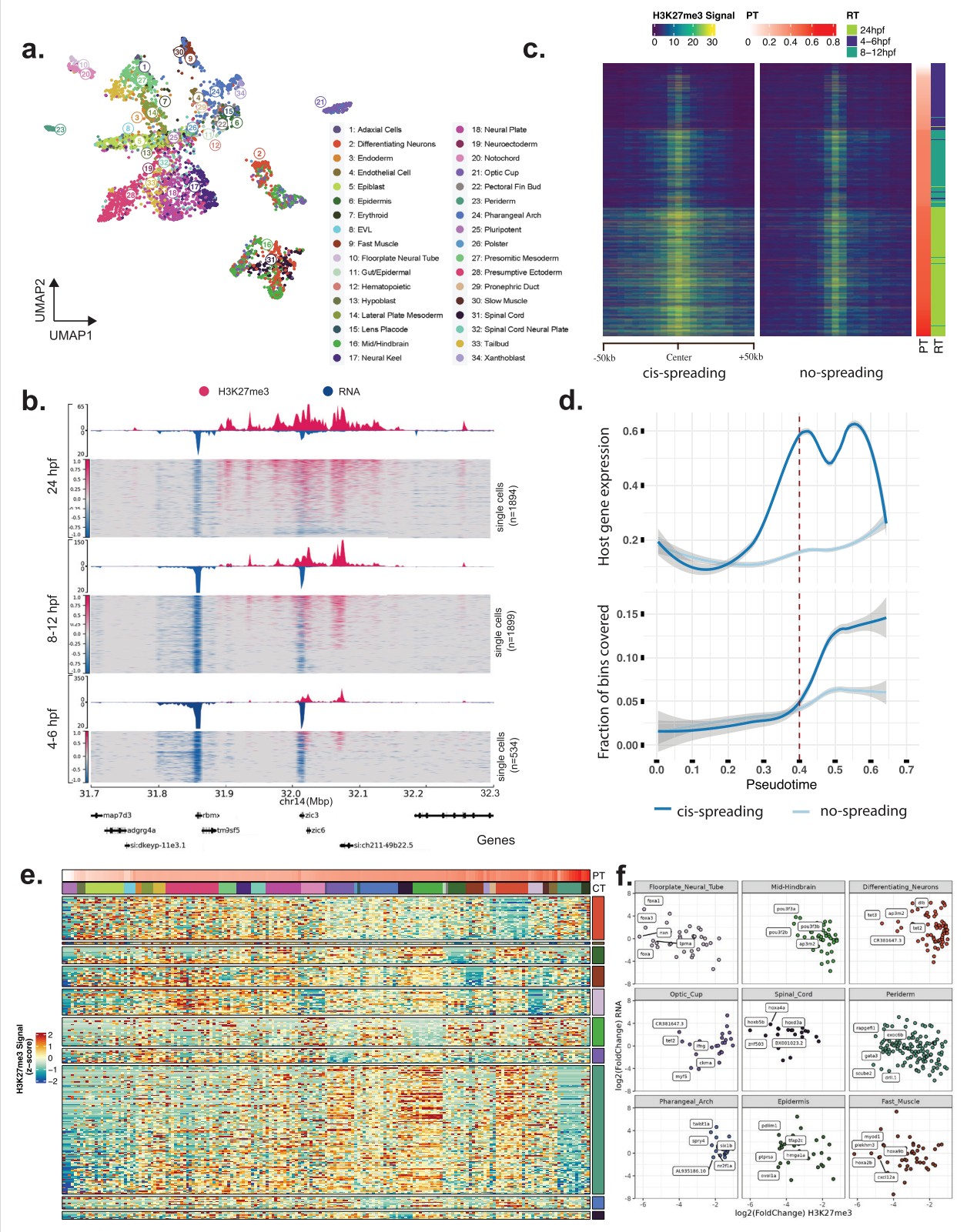

**Figure 2.** Spatio-temporal spreading of H3K27me3 correlates with gene silencing. (**a**) UMAP projection of cells based on H3K27me3 signal (same as *Figure 1b*, left) indicating the different cell types annotated using the transcriptome signal. (**b**) An example genomic locus that demonstrates the cis-spreading of H3K27me3 signal (pink) around the *zic3* gene with time during development. Note that apart from the *zic3* gene, the spreading also correlates with a downregulation of the expression (blue) for the nearby gene (*rbmx*) with time. (**c**) Single-cell heatmap. Each row is a single cell selected

*Figure 2 continued on next page*

Figure 2 continued

from ectoderm lineage (34.5% of all cells) showing the average H3K27me3 signal for the top 100 genes detected by the linear model, showing increase in spreading of signal at the 100 kb region surrounding the center bin with pseudotime. The center bin was identified as the bin with non-zero signal in pluripotent cells. PT = pseudotime, RT = real time. (**d**) Line plots comparing H3K27me3 spreading and gene expression with pseudotime. The bottom panel shows the average fraction of bins that show H3K27me3 signal on two sets of genes (spreading, non-spreading) with time, while the top panel shows the average gene expression of these gene sets along pseudotime. (**e**) Heatmap of H3K27me3 signals across cells (grouped by cell type, top) for genes with cell type-specific demethylation at 24hpf. Genes (rows) are grouped by celltype in which they are demethelyated. CT = cell type; PT = pseudotime. The analyzed cell types are indicated in the right legend, while top legend shows all cell types (colors same as **a**). (**f**) Correlation of H3K27me3 fold-change with change in gene expression, for the selected cell types. Top 5 genes with H3K27me3 loss are labeled per cell type.

The online version of this article includes the following figure supplement(s) for figure 2:

**Figure supplement 1.** Annotation of H3K27me3-RNA woT-ChIC data (4–24 hpf).

**Figure supplement 2.** H3K27me3 shows cis-spreading with time.

cell types compared to pluripotent cells, suggesting H3K27me3 loss is a cell type-specific process active during development. This loss was also proportional to a change in gene expression signal in those cell types and affected key developmental genes associated with these cells, such as POU family of transcription factors (mid/hindbrain), *tet3* and *tet2* (neurons/optic cup), *myod1* (muscle), and *tal1* (endothelium) (*Figure 2f*). This list also included many of the significant genes from our regression analysis which showed cell type-specific expression at 24 hpf, such as *gata2a*, *dlx3b*, *shha*, and *tet2* (*Figure 2—figure supplement 2i*).

Overall, our analysis shows that for a specific set of genes, silencing of gene expression is achieved once sufficient gene-body H3K27me3 coverage is achieved via cis-spreading, a process seemingly uncoupled with the prior transcription state of these genes. Further, we see that H3K27me3 demethylation can occur later in development as key developmental genes are re-activated in their corresponding cell types in a cell type-specific manner.

## Global chromatin state of cells is decoupled from gene expression during early development

Considering the heterogeneity in the repressive chromatin landscape of cells observed as early as gastrulation, we asked how the interplay between active and silenced chromatin is established at this stage. To map the active chromatin, we focused on H3K4me1, a histone modification associated with active and poised enhancers and promoters, which, unlike H3K27me3, has been observed before zygotic genome activation (ZGA) in zebrafish (*Murphy et al., 2018*). To mitigate the interference of maternally contributed RNA, we implemented a new cell preparation protocol within woT-ChIC ('Materials and methods'), which leads to the expulsion of cytoplasmic RNA from the cells (hereafter referred to as 'nuclei' batch). We generated woT-ChIC data for H3K4me1 at 4, 6, 8, 10, and 12 hpf. As expected, our nuclei dataset shows a fourfold higher ratio of unspliced RNA compared to spliced RNA, and an overall lower number of detected genes compared to the whole-cell data, in line with the expected lack of spliced maternal RNA in the nuclei (*Figure 3—figure supplement 1a and b*, *Supplementary file 2*). The chromatin quality was unaffected, exemplified by the similar number and pattern of H3K27me3 MNase cuts with time from the 'nuclei' and 'whole cell' batch (*Figure 3—figure supplement 1c*). Finally, we integrated our nuclei dataset with the 4–12 hpf subset of the whole-cell H3K27me3 woT-ChIC dataset, creating a high-quality multi-omic dataset of 15,961 cells (H3K27me3: 9197, H3K4me1: 6764) covering zebrafish gastrulation (*Figure 3a*, *Figure 3—figure supplement 1d and e*, 'Materials and methods').

With our integrated dataset, we first asked how the global chromatin state of the cells changes with time. Comparing total MNase cuts for H3K4me1 and H3K27me3 with time, we observed that while the H3K27me3 signal globally increases in cells with time, the H3K4me1 signal decreases (*Figure 3—figure supplement 1c*). To understand whether this global change stems from a change in the activity of cis-regulatory elements (CREs), we separated the data into H3K4me1 enriched regions (representing active or poised promoters and enhancers), H3K27me3-enriched regions (mostly observed near genes/promoters in earlier analysis), and other (mostly intergenic) regions. While the majority (84%) of H3K27me3-enriched regions were found to overlap with an H3K4me1 domain and show increasing H3K27me3 signal with time, this increase is not accompanied by a decrease in H3K4me1 on these regions (*Figure 3b*, *Figure 3—figure supplement 2a*). Instead, the decrease in signal was

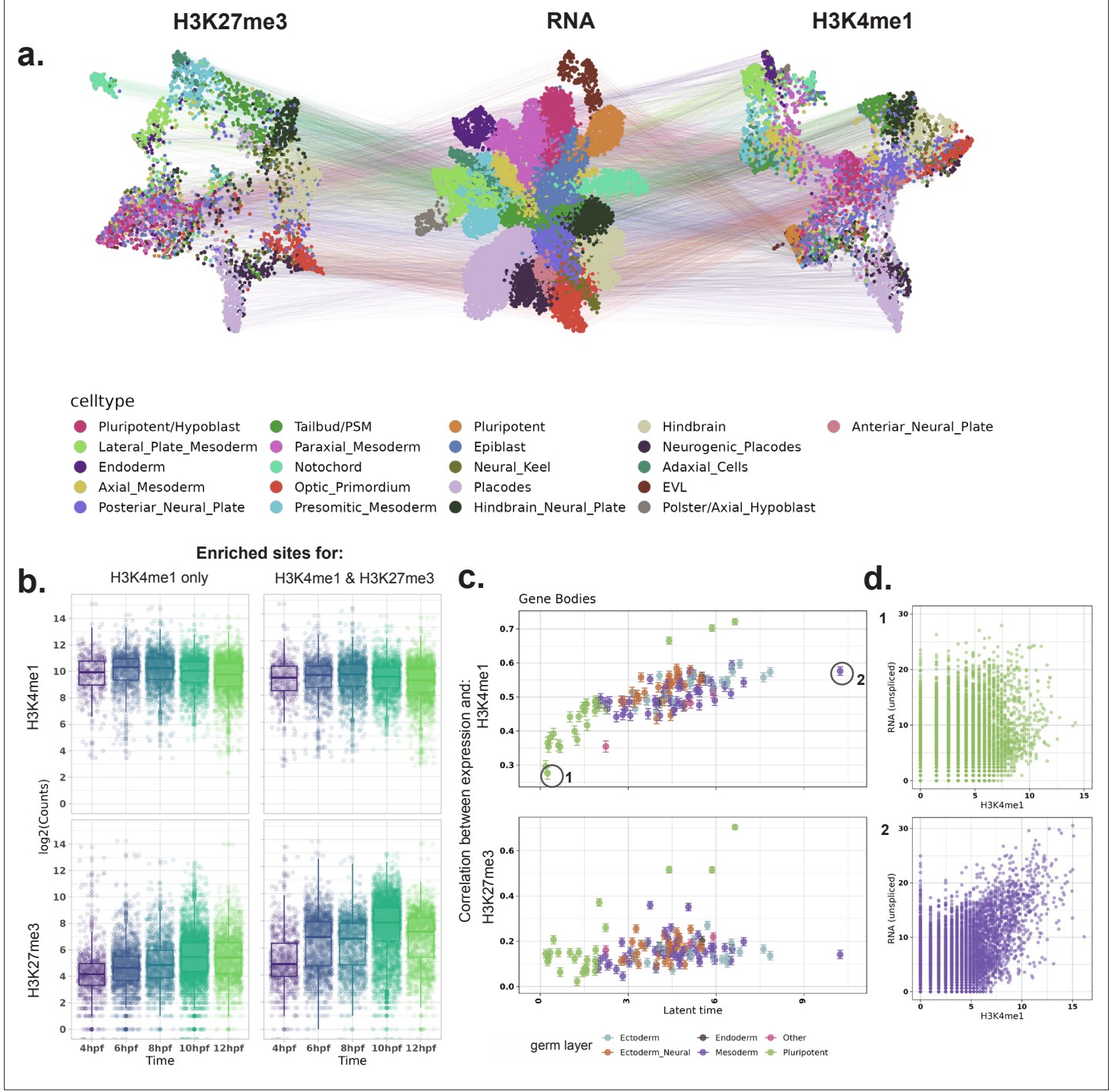

**Figure 3.** Integrative analysis of H3K27me3, H3K4me1, and transcriptome. (**a**) UMAP projection of the single cells based on the three modalities H3K27me3 (left), RNA (center), and H3K4me1 (right) after integration of the two batches and annotation of cells. (**b**) Total UMI counts per cell on H3K4me1 enriched regions from the integrated dataset, divided into H3K4me1-unique regions and regions co-enriched for H3K27me3. H3K4me1 signal (top panels) on these regions remains unchanged with time, while H3K27me3 signal (bottom panels) increases. (**c**) Correlation between histone modification signal over gene bodies and gene expression, per metacell. Metacells are ordered by latent time (X-axis) and the Pearson correlation coefficient (Y-axis) between H3K4me1 and RNA (top) and H3K27me3 and RNA (bottom). (**d**) Scatterplot showing H3K4me1 signal and unspliced RNA counts for all genes of the two selected metacells indicated in (**c**), early (1) and late (2) in latent time. Colors corresponds to the germ layer annotation of the metacell, as indicated in (**c**).

The online version of this article includes the following figure supplement(s) for figure 3:

**Figure supplement 1.** Quality control and comparison of H3K27me and H3K4me1 signal in nuclei.

*Figure 3 continued on next page*

**Figure supplement 2.** Comparison of genomic distribution of H3K4me1 and H3K27me3 with time.

**Figure supplement 3.** Derivation of latent time and lineages using RNA velocity on 4–12 hpf data.

observed in a minor fraction of H3K27me3-unique sites, and random genomic regions away from enriched sites (*Figure 3—figure supplement 2b*), suggesting that this global change in signal does not represent a change in CRE activity. Further, the ratio of H3K4me1 to H3K27me3 suggests that most promoters remain in a 'bivalent' chromatin state during 4–12 hpf in all germ layers, with a small fraction showing increased H3K4me1 activity in any specific germ layer (*Figure 3—figure supplement 2c*).

To obtain a more fine-grained view of cellular differentiation time and lineages on our integrated data, we took advantage of the high unspliced counts from our protocol. We applied the RNA velocity model (*La Manno et al., 2018*), which uses the ratio between spliced and unspliced reads of genes to obtain the cell's differentiation path and assigns a 'latent time' to the cell, indicating their differentiation stage (*Figure 3—figure supplement 3a–e*). We then asked how the change in the cell's chromatin state relates to the transcription of genes during their differentiation. For this, we aggregated transcriptionally similar cells into metacells and correlated the H3K4me1 and H3K27me3 signals with unspliced (i.e., newly transcribed) RNA signals for all genes in each metacell. Interestingly, we observed that the correlation between the gene-body H3K4me1 and transcription increases with the average latent time of a metacell (*Figure 3c and d*). Promoter regions, however, did not show this trend (*Figure 3—figure supplement 2d*). For H3K27me3, the global signal was mostly uncorrelated with transcription on both promoters and gene bodies (*Figure 3c*, *Figure 3—figure supplement 2d*). This suggests that despite increasing heterogeneity of chromatin signal, the overall chromatin state of a cell is decoupled from its transcriptional state during early development, and this coupling increases as the cells mature.

## The chromatin state of binding sites predicts the function of transcription factors during gastrulation

We next asked whether our integrated dataset could inform us about the regulation of transcription factor (TF) networks and their role in lineage specification. While many lineage-defining TFs are biochemically predicted to have both activation and silencing functions, we hypothesized that the level of H3K4me1 on transcription factor binding sites (TFBS) might indicate which function the TFs play in a cell. For example, if a TF expression is correlated to a gain in H3K4me1 on its binding sites, it could indicate its role as a transcriptional activator, while a loss in H3K4me1 on TFBS might indicate a silencing function in that cell. Further, if a TF function is epigenetically regulated, then the chromatin state of the TF itself would also be predictive of its function. Based on this idea, we built a prediction model that combines the chromatin state and expression of TFs with their H3K4me1 activity on TFBS within cells ('Materials and methods'). With a combined model, we aim to classify TFs based on both their own regulation via chromatin state (regulated/independent), as well as their action on their targets (activator/repressor) (*Figure 4a*).

Our model predicted the H3K4me1 activity at TFBS with high accuracy ($R^2 > 0.6$) for 45 TFs. Our classification captured the well-established developmental function of TFs, such as the activating function of *tbx16* in regulating paraxial mesoderm formation (*Payumo et al., 2016*), and that of *tfap2a*, a transcriptional activator shown to be important for neural crest induction (*Dooley et al., 2019*; *Figure 4b*). Further, it helped resolve the cell type-specific functions of TFs predicted to be activators or repressors based on their protein domains (*Figure 4—figure supplement 1a*, *Supplementary file 4*). For example, *zbtb16a*, predicted to have a DNA-binding transcriptional repressor activity, and *zeb1a*, speculated as a context-dependent activator/repressor (*Gheldof et al., 2012*), were both revealed as a repressor during neural ectoderm (hindbrain) specification. Next, we asked whether the gain/loss of H3K4me1 activity is reflected in the gene transcription, measured as a change in nascent (unspliced) transcripts on the genes nearest to the TFBS. For our top predicted activators and repressors, we observed the expected up and downregulation of average nascent (unspliced) RNA signal of the target genes, corresponding to the change in TF expression and activity (*Figure 4—figure supplement 1b*). This indicated that our model can capture new cell type-specific activation/

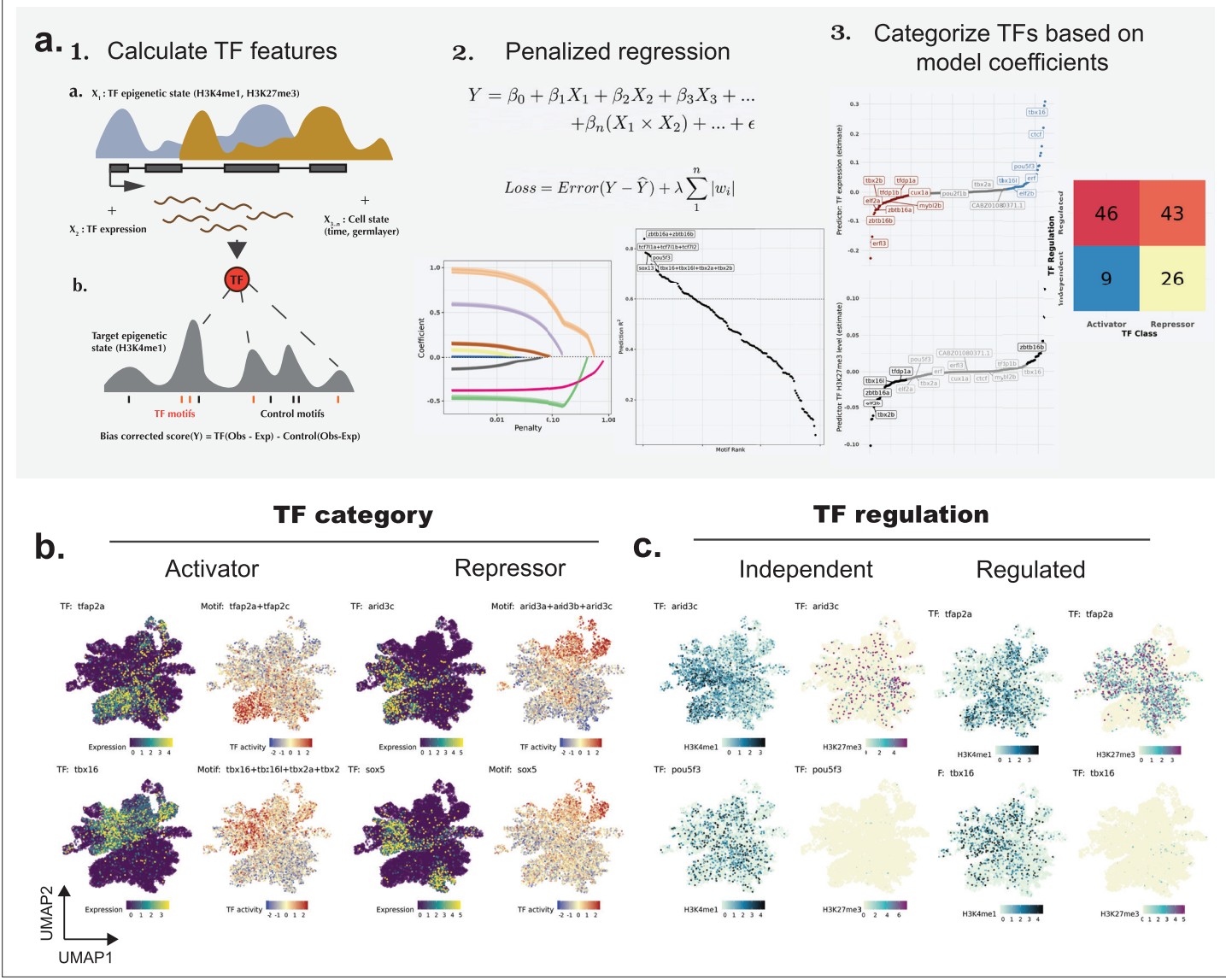

**Figure 4.** Prediction of TF activity using TF epigenetics and transcription. (**a**) Schematic and outputs of the prediction model. (1) (top) Normalized H3K4me1 and H3K27me3 signal on the TF locus, TF (spliced) RNA signal, and indicators of cell state (pseudotime and lineage) are used to predict TF activity (bottom). (2) The lasso regression model is used to select the most useful predictors for each TF. The bottom right plot shows the TFs sorted by the $R^2$ values on the independent test dataset. (3) Coefficients from the final models are ranked and compared to categorize the TFs. The number of TFs classified as activator/repressor, or regulated/independent is shown in the right plot. (**b**) UMAPs showing the expression and motif activities of TFs classified as 'activators' or 'repressors' using the above model. TF expression (left) is based on (normalized) spliced RNA and TF activity (right) is based on H3K4me1 signal on TFBS. For activators, the motif activities on TFBS are gained in cells where the TFs are expressed, while for the repressors, the motif activities are lost in the cells expressing the TF. (**c**) UMAPs show (normalized) H3K4me1 signal and H3K27me3 signal on TF gene body, for TFs classified as 'regulated' or 'independent'. The histone modifications on regulated TFs are correlated with their activities.

The online version of this article includes the following figure supplement(s) for figure 4:

**Figure supplement 1.** Examples of TFs with predicted activation and repression functions and their epigenetic regulation.

repression functions of TFs during gastrulation. Additionally, our model also detects TFs that are epigenetically regulated (*Figure 4c*, *Figure 4—figure supplement 1c*, *Supplementary file 4*). For TFs such as *sox13, tbx16, lhx1a, tfap2a*, a gain of H3K4me1, or a loss of H3K27me3, or a combination of both was associated with their respective TFBS activity. This indicates that while the majority of TFs expressed early in development appear to be regulated by alternative mechanisms, the chromatin state could play an important role in establishing the gene expression memory for a subset of developmentally important TFs.

# Discussion

In this study, we adopted our recently developed T-ChIC method (*Zeller et al., 2024*), to study the dynamics of active (H3K4me1) and silencing (H3K27me3) histone modifications during early zebrafish development. We observe a dynamic spatiotemporal localization of these histone modifications, previously unresolved by bulk chromatin profiling assays. This data allows for a direct comparison of the chromatin state and the expression of genes in individual cells and helps to understand the role of this interaction in regulating cell fates during early development.

We observe that H3K27me3 shows a promoter-anchored spread during development. At the start of differentiation, selected genomic loci with multiple promoters are pre-marked with a broadly distributed H3K27me3 (such as the *hox* and *pcdh1* loci), while loci with single promoter (such as *pax7b*) appear as focused H3K27me3 domains. A recent study has shown that such pre-marking is established by a non-canonical interaction between the two polycomb (PcG) complexes, PRC1 and PRC2 (*Hickey et al., 2022*). Here, we find that a selected set of these loci shows the spreading of H3K27me3 with differentiation, which eventually confers the silencing of host genes. A recent study using mouse embryonic stem cells proposes nucleation and spreading as a way to maintain PcG silencing (*Veronezi and Ramachandran, 2024*). Based on our observations, we propose that this mechanism could also help to propagate the spread of silencing during development. While this spread appears to be mostly not lineage-specific, we do observe a cell type-specific demethylation of many genes which are developmentally important for cell type specification, suggesting that the regulation of developmental genes via H3K27me3 could be established through a lineage-agnostic spread, followed by cell type-specific demethylation. This could likely be a fundamental mechanism to establish a cell type-specific gene expression memory via the polycomb complex considering the silencing of developmental genes in 'alternate' lineages is a conserved function of H3K27me3 (*Guo et al., 2021*). We see that in the absence of H3K27 demethylation, important developmental genes such as *hox*, *pax*, and *shh* genes are silenced in a spatiotemporal manner. A mislocalized expression of these known PcG targets has been observed after a deletion of the core PRC2 enzyme, *ezh2* (*San et al., 2016*; *Yette et al., 2021*). Apart from confirming these known targets, we additionally identify developmental genes such as *rfx4*, important for neural tube formation (*Sedykh et al., 2018*), *dlx3b*, important for placode development (*Esterberg and Fritz, 2009*), among others, as novel PcG targets.

Although a rather large number of gene promoters are marked by H3K27me3 in pluripotent cells, only a minority of these show a genomic spread and silencing of the genes. By comparing the H3K4me3 to H3K27me3 signal on promoters, we find that most of these promoters are co-marked at 4–12 hpf, suggesting that they might serve as a 'placeholder' for activation or silencing later in development. This provides an explanation to why the cis-spreading, but not the promoter enrichment of H3K27me3 is linked to gene silencing during development.

In line with previous studies (*Kaaij et al., 2016*; *Murphy et al., 2018*), we find that H3K4me1 is widespread in the genome of pluripotent cells, marking a large number of TF motifs and other genomic regions. While this chromatin mark systematically disappears in regions outside of cis-regulatory elements (CREs) during development, its activity on the CREs does not show a monotonous change with time. In fact, a systematic increase in H3K27me3 without a loss of H3K4me1 leads to a bivalent chromatin state on most CREs, together with a lineage-specific gain or loss of H3K4me1 on selected CREs. We show that these changes in H3K4me1 levels can be leveraged to predict the lineage-specific activator or repressor functions of TFs, by correlating this activity with the TF's own expression and chromatin states. Using this approach, we find novel functions of TFs in lineage specification, such as the role of *zbtb16a/b*, *zeb1a/b* as negative regulators during ectoderm specification, and the *tfap2a/b* as a positive driver of non-neural ectoderm during gastrulation. We also find selected lineage-specifying TFs such as *zfhx3*, *foxc1a*, and *irx3a* whose activity seems to be regulated by their own chromatin state during gastrulation. These results might point to a new pathway through which the chromatin states of the cells play a role in specifying cell fates, that is, by establishing a transcriptional memory on key lineage regulators.

Overall, comparing the active and silenced chromatin states of cells, we observe that the active state is a better predictor of a cell's functional (transcriptomic) state in early development. A caveat is that we have not mapped other important silencing chromatin states, such as H3K9me3 or DNA methylation, which may show complementary dynamics in early development. We also see that both active and silencing states are rather uncoupled from transcription in pluripotent cells and get correlated

as the cells mature in development. Note that this maturation time is not necessarily the same as the developmental time (hpf) of the embryo, as the transcriptionally mature cells collected from early time points also show a high correlation of active chromatin state and transcription. Therefore, we propose that a correlation of chromatin and transcriptional state of cells could be a hallmark of cell identity formation during development. Future studies to systematically map the overall chromatin state of single cells and gene expression would further explain how cell fates are established during embryogenesis.

## Materials and methods
### Whole-organism T-ChIC of zebrafish embryos

The detailed, step-by-step woT-ChIC protocol of zebrafish embryos (from embryo collection to the preparation of sequencing libraries) is available at: https://dx.doi.org/10.17504/protocols.io. q26g7pbe8gwz/v2. Below, we briefly describe the cell collection and staining steps used to produce this dataset.

Wild-type TL embryos were collected 20 minutes after fertilization in a Petri dish with E3 medium and kept at 28.5°C in an incubator. During the first hour, the unfertilized embryos were discarded. At the desired stage, embryos were dechorionated by incubation in 1 mg/mL of pronase, and 30–50 embryos were deyolked in Ca-free Ringer's solution, pelleted, and washed with 500 µL of PBS + 10% FBS. For early time points (4, 6, and 8 hpf), cells were dissociated with the addition of 200 µL of pre-warmed FACSmax cell dissociation solution (Genlantis T200100) for 5 minutes resuspending gently up and down at room temperature (RT). For later time points (10, 12, and 24 hpf), cells were dissociated with the addition of 200 µL of pre-warmed Protease solution for 6 minutes on a shaker at 28°C and 400 rpm, resuspending gently every 2 minutes. After dissociation, cells were filtered with a 35 µL sieve (Corning, 352235) and washed with 500 µL of PBS + 10% FBS and resuspended in Wash Buffer 1 (WB1, described in the online protocol) and kept at +4°C before starting the CellTracer staining. For the 'nuclei' batch, WB1 was modified with 0.05% Saponin (Sigma, 47036-250G-F) instead of 0.05% Tween20. Cells were vortexed well and kept in the dark at +4°C for 20 minutes to stain with a combination of CellTrace dyes (Thermo Fisher C34570, Thermo Fisher C34573, Thermo Fisher C3457, and a combination of two of these). The staining was stopped with the addition of 70 µL of rat serum (Sigma-Aldrich, R9759-5ML) and a 5-minute incubation at RT. Lastly, cells were washed and resuspended in WB1 with spermidine solution (0.072 µL/mL) and 4 µL/mL 0.5 M EDTA. Once all time points had been stained with their appropriate dye/dyes combinations, they were pooled in a 0.5 mL protein-low binding tube with approximately 1 million cells in total. Cells were incubated overnight at 4°C with primary antibodies (1:200 H3K27me3 rabbit mAB, Cell Signalling #9733; 1:100 H3K4me1 polyclonal Ab, Thermo Fisher #710795). The next day, the cells are washed and incubated with pA-M-Nase in WB1 for 1 hour, washed, and sorted into indexed 384-well plates containing CelSeq2 adapters. Cells were incubated for 30 minutes at 4°C for MNase digestion and stopped with the stop solution before proceeding with the rest of the library preparation steps. The pA-MNase fusion protein was produced as described earlier (*Schmid et al., 2004*). Following T-ChIC library preparation and QC, the final DNA libraries are sequenced paired-end 100 bp, on either a NovaSeq or NextSeq2000, at a sequencing depth between 15 and 25 million reads per sample (384-well plate).

### Processing and quality control of T-ChIC data

The first-in-pair reads from the T-ChIC protocol contain an RNA or ChIC barcode in the following format "*RNA: 6N7X, ChIC: 3*N8X"; where N=UMI nucleotide and X=Cell barcode nucleotide. We used a custom Python script to split the raw *.fastq* files into the ChIC and RNA fractions based on which one of the two barcode patterns is observed at the start. The two fractions are then independently mapped to the GRCz11/danRer11 genome. A complete processing workflow (from .fastq to count tables) with all parameters is available at https://github.com/bhardwaj-lab/scChICflow, copy archived at *Bhardwaj and Sancho Gómez, 2025* (v 0.4) and is briefly described below.

The **RNA fraction** was trimmed using cutadapt (v2.1) (*Martin, 2011*) with parameters `-e 0.1 -q 16 -O 3 --trim-n --minimum-length 10 --nextseq-trim=16 A W{'10'}`, along with Illumina truseq barcodes provided as `-a and -b` options. The trimmed reads are mapped to the genome using STAR (v 2.7.11) (*Dobin et al., 2013*), using the "*StarSolo*" mode, with these important

parameters `--sjdbGTFfile <dr11_ens104.gtf> --outFilterIntronMotifs RemoveNoncanonical --soloCBmatchWLtype Exact --soloType CB_UMI_Simple`, where "*dr11_ens104.gtf*" refers to the ENSEMBL annotation version 104 (GRCz11) (*Cunningham et al., 2022*). Secondary and supplementary alignments and low-quality mappings (<MAPQ 255) were removed using samtools (v1.21) (*Li et al., 2009*) and reads were de-duplicated with UMI-tools (v.1.0.0) (*Smith et al., 2017*) using cell barcode and UMI position, along with options `--method unique --spliced-is-unique`. Coverage files were created using deepTools *bamCoverage* with CPM normalization (*Ramírez et al., 2016*).

For the **ChIC fraction**, barcodes were moved into the read header using UMI-tools *extract* (v.1.0.0). Reads were trimmed using cutadapt (v2.1) with parameters `-e 0.1 -O 5 -u 1 -u -2 -U -2 -a W{10} -A W{10} -q 30 --trim-n --minimum-length 20 --nextseq-trim=30`, along with illumina truseq barcodes provided as `-a and -b ` options. The trimmed reads are mapped to the genome using hisat2 (v2.2.1) (*Kim et al., 2017*), with parameters `--sensitive --no-spliced-alignment --no-mixed --no-discordant --no-softclip -X 1000`. Reads were de-duplicated with UMI-tools (v.1.0.0) using cell barcode and UMI position, along with options `--method unique --spliced-is-unique --soft-clip-threshold 2`. Quality control was performed using deepTools. Reads were counted on 50 kb windows in the genome using sincei (*Bhardwaj and Mourragui, 2024*).

## Analysis of publicly available data

We downloaded the raw *.fastq* files for 6 hpf bulk CUTnRUN data of H3K27me3 from Akdogan-Ozdilek et al. (GSE178343) (*Akdogan-Ozdilek et al., 2022*), and raw *.fastq* files of 12 hpf and 24 hpf ChIP-seq data from the danio-code portal (accessions - 12 hpf H3K27me3: DCD003854SQ, 12 hpf H3K4me1: DCD003854SQ, 24 hpf H3K27me3: DCD003200SQ). All *.fastq* files were mapped to the GRCz11 genome using snakePipes' DNA-mapping workflow, with parameters `--trim --fastqc --mapq 5 --dedup --bwBinSize 1000` (*Bhardwaj et al., 2019*). The de-duplicated BAM files were subsampled to match the sequencing depth of the corresponding pooled time points (early vs 6 hpf, middle vs 12 hpf, late vs 24 hpf), and the read coverage was compared using deepTools (*Ramírez et al., 2016*) multiBAMSummary (with parameter `-bs 50000`) and plotFingerPrint (with parameters `--skipZeros -bs 10,000-n 50000`). For the analysis of H3K27me3 signal in mouse embryos, we downloaded the H3K27me3 bedgraph files corresponding to stages: E5.5, Endoderm, Ectoderm, and Mesoderm by *Xiang et al., 2020* (GSE125318), and plotted the signal over the +50 kb bins surrounding the mouse gene transcription start sites (mm9 genome) using deeptools ComputeMatrix (with additional parameters `-bs 500 --missingDataAsZero --skipZeros --maxThreshold 1000`) and plotHeatmap.

## Cell clustering and annotation using RNA signal

For clustering of single cells based on RNA signal, we used the 'spliced' count matrices for 'whole-cell' T-ChIC data, and 'total' (spliced + unspliced + ambiguous) counts for 'nuclei' T-ChIC data. Filtering and clustering of cells were performed in scanpy (v1.9.1) (*Wolf et al., 2018*). We removed cells with `total_counts <1000, or n_genes_by_counts >10000, or pct_counts_in_top_100_genes >0.6`. We also removed cells with <70% of counts on protein-coding genes. We selected genes present in at least 1% of cells (or at least 50 cells, whichever is smaller) and selected the top 4000 variable genes based on their analytical Pearson residuals (*Lause et al., 2021*). We used the Pearson residuals to calculate principal components (PCs) and built a neighbor graph using 50 PCs and 30 neighbors (20 for nuclei data). We then used it to build a paga graph (*Wolf et al., 2019*) based on Leiden clusters (*paga threshold = 0.1, leiden resolution = 1.5*). For 2D representation, UMAPs were initiated with the paga graph, along with additional parameters `min_dist = 1, spread = 5` (*spread = 1* for nuclei).

For the annotation of cell types and all other analyses, we calculated the normalized ChIC and RNA signal using the 'shifted log transform' method (`1/sqrt(alpha) log(4 * alpha * x+1)`) (*Ahlmann-Eltze and Huber, 2023*), with a fixed overdispersion (alpha) of 0.05 and total counts ("normed_sum") as library size factors. For annotation of cells, we obtained the raw count matrices from *Wagner et al., 2018* and subsetted for the 4, 6, 8, 10, 14, and 24 hpf timepoints (for the 'nuclei'' batch, we also excluded 24 hpf). We normalized the counts in the same manner as our counts

and selected the top 4000 variable genes (using `FindVariableFeatures(selection.method = "vst")` in seurat). We then performed CCA-MNN analysis in Seurat using `FindTransfer-Anchors(method="cca")` and used the transfer score to predict labels for single cells. For top predicted labels for each cluster, we then manually confirmed the marker gene expression from ZFIN in the respective cluster, followed by renaming the cluster to suitable ZFIN cell ontology (*Bradford et al., 2022*).

## Integrated analysis of nuclei and whole-cell data

To integrate the 'nuclei'' and 'whole-cell' subsets of data, we merged the cells from the 'nuclei'' batch with that of 4–12 hpf subset of the 'whole-cell' batch, resulting in 15,961 cells. We then removed genes with total spliced or unspliced counts <100, or genes detected in <100 cells, from the merged data, and calculated the top 4000 variable genes (HVGs) based on their analytical Pearson residuals (*Lause et al., 2021*) and used the intersection of the HVGs from the two batches (3091 genes) to perform PCA based on the Pearson residuals of the 'unspliced' counts from the two batches. Top 50 PCs were then used to align the two batches using harmony (*Korsunsky et al., 2019*). The harmony-corrected PCs were used for further analysis of the joint dataset (clustering, annotation, metacells, and RNA velocity).

For the calculation of latent time and lineages of cells on the integrated 4–12 hpf data, we combined the RNA velocity and diffusion pseudotime approach, using cellrank (v1.5.1) (*Lange et al., 2022*). We calculated RNA-velocity using the 'dynamical' model as described in the scVelo package (v0.2.4) (*Bergen et al., 2020*). The cell-specific moments *Ms* and *Mu* were calculated using the HVGs and PCs from the above analysis, and top 1000 genes were used for the dynamical model to calculate the gene-shared latent time and cell-specific velocities (`scv.tl.recover_dynamics` with parameters: `fit_connected_states = False, max_iter = 50, t_max = 12, fit_basal_transcription = True`, `scv.tl.velocity` with parameters: `min_r2=0.2, groups_for_fit = <8-12 hpf>`). The latent time was combined with connectivities using cellrank (`cr.tl.transition_matrix` parameter: `weight_connectivities = 0.2`), and one initial and six terminal states were calculated using cellrank's GPCCA estimator.

## Metacell analysis

To obtain a detailed view of cellular heterogeneity while reducing dropouts, we aggregated transcriptionally similar cells into the so-called 'metacells', based on archetype analysis implemented in the SEACells python package (*Persad et al., 2023*). We used SEAcells with parameters `n_SEACells = nc, n_waypoint_eigs = 15, convergence_epsilon = 1e-5`, where nc = 180 for whole-cell data and nc = 160 for the integrated 4–12 hpf data. Each metacell was then annotated with the mean latent time or pseudotime of underlying single-cells, and the max number of cells belonging to an annotated cell type or collection time (hpf). For analysis involving a comparison of H3K4me1 and H3K27me3, the underlying number of single cells was downsampled for each metacell, such that equal number of cells from both the histone modifications were assigned to each metacell, and only metacells with a minimum of 20 cells from both histone modifications were kept, to assure robust results.

## Cell clustering using the ChIC signal

For the clustering of single cells based on ChIC signal, we performed latent semantic analysis (LSA) using the gensim package in python (*Řehůřek and Sojka, 2010*). The `Cells*Regions` sparse matrix was treated as a vector of documents (Cells), where region counts represent word frequency. The documents were then transformed with log term-frequency (TF), inverse document frequency (IDF) as follows:

$$tf_{td} = 1 + \log_2 f_{i_k}$$

$$idf(t, D) = \log_2 \left( \frac{N}{n_k} \right)$$

$$TF - IDF(t, d, D) = tf_t, d * idf_t, D$$

Where $f_ik$ refers to the count frequency of a (50 kb) genomic bin $T_k$ in a cell, $D_i$ and $nk$ refer to the number of cells containing non-zero counts for the bin. $T_k$ The output is subjected to a pivoted unique normalization (*Singhal et al., 2017*) to take into account the difference in total number of detected regions per cell.

$$pivotednorm = (1.0 - slope) * pivot + slope * TF - IDF(t, d, D)$$

In our case, we calculated pivot as the average number of non-zero bins across all cells, and fixed the slope to *0.25* (recommended by *Singhal et al., 2017*). The resulting matrix is subjected to a truncated SVD (singular value decomposition) (*Halko et al., 2009*), yielding `Cell*Topic` and `Region*Topic` matrices. Similar to scRNA-seq, we calculated 30 nearest neighbors using the 50 topics from the LSA output (dropping Topic-1, which strongly correlates with read depth), and used it to build the paga graph (with *threshold = 0.1*). UMAPs were initialized using the PAGA graph, with parameters `min_dist = 0.1, spread = 5`. Leiden clusters were calculated on the neighborhood graph with `resolution = 1.5`.

## Peak calling and annotation

For the detection of regions with both narrow and broad enrichment in the genome, we used a two-step peak calling approach. We first pooled all our filtered cells from 4 to 24-hour time points into a BAM file and used histoneHMM (v1.7) function `call_regions` with parameters `-bs 750 P 0.1` (*Heinig et al., 2015*), and further removed the detected regions with average posterior probability <0.4, and referred to them as 'domains'. Next, we performed peak-calling using MACS2 (v2.2.4) (*Zhang et al., 2008*) on the same file, with parameters `--mfold 0 50 --extsize 200 --broad --keep-dup all`, and overlapped these peaks with the domains detected from histoneHMM. Peaks overlapping with the histoneHMM domains, and having a local enrichment score ≥ 50, were referred to as 'subpeaks'. For further analysis, we replaced the domains containing subpeaks with their respective subpeaks, resulting in 11221 enriched domains for H3K27me3 and 74,004 domains for H3K4me1.

For peak annotation, we used the *genomicRanges* R package (*Lawrence et al., 2013*) to classify these domains into 'promoter' (within +300 or –200 bases of a transcription start site), genic (overlapping a gene body, but not promoter), and 'intergenic' (outside promoters or gene body). The 'genic' domains were reclassified into 'gene covering' if they covered ≥ 80% of a gene. All domains were annotated with the gene(s) that overlapped with or (in the case of intergenic domains) were nearest to them. To detect which cis-regulatory elements are present inside these domains, we overlapped them with the location of 'consensus PADREs' annotated by the danio-code project (*Baranasic et al., 2022*). We used the gimmemotifs (*van Heeringen and Veenstra, 2011*) (v0.18.0), with parameters `scan -N 30` to annotate these peaks for the associated transcription factor binding sites (TFBS), using the `vertebrate.v5.0` motif database. The detected motifs were then filtered for zebrafish TF motifs that are also present in the SwissRegulon (dr11) database (*Pachkov et al., 2007*), resulting in 590 motifs belonging to 912 TFs.

## H3K27me3 spreading and demethylation analysis

To detect the regions in the genome showing H3K27me3 cis-spreading, we used the table of 5 kb bin counts in single cells. We defined the 'center' bin as the bins showing non-zero counts in ≥ 5% of 'pluripotent' cells (784 bins), and 'neighbor' bins as the 10 up and downstream bins to the center bins. We then applied linear regression to predict the counts in 'neighbor' bins ($\hat{Y}$) as a function of the sum of counts in the "center" bins ($\hat{X}$) across metacells.

$$\hat{Y}_i = \hat{\beta}_0 + \hat{\beta}_1 X_i + \hat{\epsilon}_i$$

To test if the spread is germ layer-specific, we compared this model to a second model including germ layer covariate ($\hat{X}_2$) via a likelihood ratio test.

$$\hat{Y}_i = \hat{\beta}_0 + \hat{\beta}_1 X_i + \hat{\beta}_2 X_2 + \hat{\epsilon}_i$$

Similarly, we used a linear regression model formulation above for the prediction of H3K27me3 silenced genes, where

$$\hat{Y} = log2((sum(counts_{un}) * sum(introns_{un})/1000))$$

and

$$\hat{X} = log2(counts_{k27} * length_{k27}/1000)$$

$counts_{un}$ represent unspliced counts of genes overlapping a H3K27me3 peak, and $introns_{un}$ is the length of their introns. $counts_{k27}$ are ChIC counts on the H3K27me3 domain and $length_{k27}$ is the length of the domain.

For the analysis of loss of H3K27me3 upon differentiation. We assigned cell types to meta-cells based on the highest proportion of cell labels in that group, then summed up the gene-level H3K27me3 counts. We then filtered the cell types with <3 metacells and used edgeR (**Robinson et al., 2010**) to perform a differential signal analysis between cell types using metacells as biological replicates. We used the following workflow: `filterByExpr(dge, min.count=5, min.total.count= 20, min.prop=0.3); calcNormFactors(), estimateDisp(design), glmQLFit(dge, design)` with default parameters. Next, we filtered genes with `FDR <0.05, logFC < −1` to only retain genes with a loss of H3K27me3. We perform the same analysis for spliced and unspliced RNA counts for these genes (except filtering by logFC) to compare their results with that of H3K27me3 in the same cell types.

## TF activity prediction and classification

To calculate the TF activity using H3K4me1 signal, we filtered our annotated H3K4me1 peaks (with assigned cPADREs) for peaks uniquely enriched for H3K4me1. Zebrafish TF motifs from the Swiss-Regulon (dr11) database (**Pachkov et al., 2007**) (590 motifs belonging to 912 TFs) were assigned to the peaks, based on motif match score inside the cPADREs within those peaks. Next, we obtained the H3K4me1 counts per peak per cell and assigned these counts to each TF motifs annotated with these peaks. Finally, we converted these raw counts into bias-corrected TF motif deviance scores per cell using chromVar (**Schep et al., 2017**). We also calculated deviance scores for metacells, using the aggregated counts per metacell, instead of single cells.

For TF activity prediction, we calculated the normalized H3K4me1 and H3K27me3 and (spliced) RNA counts per metacell, along with the metacell annotations (germ layer, latent time) and used them to predict the TF activity using lasso-penalized regression (**Tibshirani, 1996**). We first divided the data into a 70–30 (training/test) set and used the training set to tune the penalty parameter ($\lambda$) using grid search on 10-fold resamples. The top model was then run on each TF separately on the test set and evaluated based on $R^2$ estimates. The $R^2$ estimates were then compared to the permutation-based estimates to obtain a significance score (p-value) and adjusted for multiple testing via Benjamini–Hochberg (BH) correction to select top TFs for classification (Padj < 0.01). For the classification of TFs, we extracted the weights for the final model for each TF at the highest, $\lambda$, and interpreted them based on previous knowledge about these histone modifications. For example, since H3K4me1 activity represents active or poised enhancer, a positive correlation (weight >0) between a TF expression and H3K4me1 activity suggests its action as an activator, while a negative correlation (weight <0) suggests a repressor. Similarly, a non-zero weight for a TF's H3K4me1 or H3K27me3 level would indicate that a TF activity is regulated by either, or both of the marks.

## Code availability

The code for processing of T-ChIC data from raw (fastq) files up to count tables is available open source with GPLv3.0 license at https://github.com/bhardwaj-lab/scChICflow (**Bhardwaj and Sancho Gómez, 2025**). The config files containing all preprocessing parameters for scChICflow, together with scripts to reproduce our figures, are available open source with CC 4.0 license on Zenodo: https://doi.org/10.5281/zenodo.16813408.

## Materials availability

Reagents, antibodies, and oligonucleotides used for our experiments are available from commercial providers (see our online protocol for a full list: https://dx.doi.org/10.17504/protocols.io.q26g7pbe8gwz/v2).

# Acknowledgements
We acknowledge the Utrecht Sequencing Facility (USEQ) for providing sequencing service and data. USEQ is subsidized by the UMC Utrecht and The Netherlands X-omics Initiative (NWO project 184.034.019). We thank Reinier van der Linden (Hubrecht FACS facility) for performing single-cell sorting. This work was supported by European Research Council Advanced under grant ERC-AdG 101053581-scTranslatomics, and the NWO consortium grant OCENW.GROOT.2019.017 to AvO. The SNF (P2BSP3-174991), HFSP (LT000209/2018-L), and Marie Skłodowska-Curie Actions (798573) supported PZ. EMBO LTF (ALTF 1197–2019) supported VB.

# Additional information

### Funding

| Funder | Grant reference number | Author |
|---|---|---|
| Nederlandse Organisatie voor Wetenschappelijk Onderzoek | OCENW.GROOT.2019.017 | Alexander van Oudenaarden |
| European Molecular Biology Organization | ALTF 1197-2019 | Vivek Bhardwaj |
| Human Frontier Science Program | LT000209/2018-L | Peter Zeller |
| H2020 Marie Skłodowska-Curie Actions | 798573 | Peter Zeller |
| European Research Council | ERC-AdG 101053581 | Alexander van Oudenaarden |
| Swiss National Science Foundation | P2BSP3-174991 | Peter Zeller |

The funders had no role in study design, data collection and interpretation, or the decision to submit the work for publication.

### Author contributions
Vivek Bhardwaj, Conceptualization, Resources, Data curation, Software, Formal analysis, Supervision, Investigation, Visualization, Methodology, Writing – original draft, Project administration, Writing – review and editing; Alberto Griffa, Data curation, Validation, Investigation, Writing – original draft, Writing – review and editing; Helena Viñas Gaza, Data curation, Investigation, Visualization, Writing – original draft; Peter Zeller, Conceptualization, Resources, Methodology, Writing – original draft; Alexander van Oudenaarden, Resources, Supervision, Funding acquisition, Writing – original draft, Project administration, Writing – review and editing

### Author ORCIDs
Vivek Bhardwaj https://orcid.org/0000-0002-5570-9338
Alberto Griffa https://orcid.org/0000-0002-0651-1173
Peter Zeller https://orcid.org/0000-0003-2537-0254

Reviewer #1 (Public review): https://doi.org/10.7554/eLife.110400.2.sa1
Reviewer #2 (Public review): https://doi.org/10.7554/eLife.110400.2.sa2
Author response https://doi.org/10.7554/eLife.110400.2.sa3

# Additional files

### Supplementary files
Supplementary file 1. Number of cells acquired per timepoint, mark and batch.
Supplementary file 2. Comparison of median counts and detected genes per cell.

Supplementary file 3. Genes with differential H3K27me3 signal between selected 24hpf cell types and others.

Supplementary file 4. Classification of TFs based on the results of the penalized regression model.

MDAR checklist

## Data availability

Raw sequencing data (.fastq), count tables (.h5ad/anndata format) with gene and cell-level metadata (including annotations) from this study are publicly available at GEO (GSE265874). Additionally, the GEO repository also contains signal tracks (.bigwigs) and peaks (.bed) files. Other source data behind our figures are available on Zenodo: https://doi.org/10.5281/zenodo.16813408.

The following datasets were generated:

| Author(s) | Year | Dataset title | Dataset URL | Database and Identifier |
|---|---|---|---|---|
| Bhardwaj V, Viñas Gaza H, Griffa A, Zeller P, van Oudenaarden A | 2025 | Single-cell multi-omic dataset mapping chromatin modifications and transcriptome during zebrafish development | https://www.ncbi.nlm.nih.gov/geo/query/acc.cgi?acc=GSE265874 | NCBI Gene Expression Omnibus, GSE265874 |
| Bhardwaj V | 2025 | Replication package for: Single-cell multi-omic analysis reveals principles of transcription-chromatin interaction during embryogenesis | https://doi.org/10.5281/zenodo.16813408 | Zenodo, 10.5281/zenodo.16813408 |

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
