## [Editor Report · eLife Assessment]

In this **valuable** study, the authors examine transcription and chromatin dynamics during early zebrafish development by simultaneously profiling histone modifications and full-length transcriptomes in thousands of single cells, providing **solid** analysis that chromatin and transcriptional states are initially weakly correlated in early embryonic cells and become progressively more aligned as differentiation proceeds. The work also supports a model in which promoter-anchored cis-spreading of H3K27me3 contributes to stable gene silencing during development. Future functional perturbations and orthogonal validations will be needed to determine the causal contribution of Polycomb spreading to fate commitment. Overall, the dataset and accompanying analyses provide a robust resource and a quantitative framework for studying chromatin-transcription relationships during vertebrate embryogenesis.

[Editors note: this paper was reviewed by Review Commons.]

---

## [Referee Report · Reviewer #1 (Public review)]

This manuscript presents a comprehensive and technically impressive study investigating the interplay between active (H3K4me1) and silencing (H3K27me3) chromatin states and gene expression during early zebrafish development. By applying an optimized single-cell multi-omics method (whole-organism T-ChIC) to profile histone modifications and transcriptomes simultaneously in thousands of cells from 4 to 24 hours post-fertilization, the work addresses a significant gap in understanding how epigenetic states are established and propagated during vertebrate embryogenesis.

There are several obvious strengths:

(1) Innovative Methodology: The adaptation and application of the T-ChIC protocol to a whole-organism, multiplexed time-course design is a major technical achievement. The generation of a high-quality, paired chromatin (H3K27me3 and H3K4me1) and full-length transcriptome dataset from the same single cells is a powerful resource for the field.

(2) Novel Biological Insights:

(2.1) It provides single-cell evidence for the promoter-anchored cis-spreading of H3K27me3 as a mechanism for gene silencing during differentiation, a process that appears largely lineage-agnostic.

(2.2) It demonstrates that global chromatin states (both active and repressive) are initially decoupled from transcriptional output in pluripotent cells and become correlated as cells mature, suggesting this coupling is a hallmark of identity formation.

(2.3) It develops a predictive model using TF expression and the H3K4me1 state at TF binding sites to infer lineage-specific activator/repressor functions and epigenetic regulation of TFs themselves, revealing novel roles for factors like zbtb16a and zeb1a.

There are also several weaknesses for further clarification:

(1) The study focuses on H3K27me3 and H3K4me1. Why these two specific histone modifications were chosen as the primary focus for this study on early fate commitment?

(2) There are some similar single-cell techniques available (histone modifications and transcription from the same single cell), what is the performance of T-ChIC when comparing to other methods？

Comments on revised version:

Other histone modifications and TFs, or even DNA methylation could be tested to see the robustness of T-ChIC.

---

## [Referee Report · Reviewer #2 (Public review)]

Summary:

Joint analysis of multiple modalities in single cells will provide a comprehensive view of cell fate states. In this manuscript, Bhardwaj et al developed a single-cell multi-omics assay, T-ChIC, to simultaneously capture histone modifications and the full-length transcriptome and applied the method to early embryos of zebrafish. The authors observed a decoupled relationship between the chromatin modifications and gene expression at early developmental stages. The correlation becomes stronger as development proceeds, as genes are silenced by the cis-spreading of the repressive marker H3k27me3. Overall, the work is well performed, and the results are meaningful and interesting to readers in the epigenomic and embryonic development fields.

Strengths:

This work utilized a new single-cell multi-omics method and generated abundant epigenomics and transcriptomics datasets for cells covering multiple key developmental stages of zebrafish.

Weaknesses:

The data analysis was superficial and mainly focused on the correspondence between the two modalities. The discussion of developmental biology was limited.

Overall, the T-ChIC method is efficient and user-friendly, and the single-cell datasets for zebrafish early development are also valuable. Audiences in the field of epigenomic and embryonic development will benefit from this work.

Comments on revised version:

The authors have answered my previous concerns.

---

## [Author Response]

**General Statements**

We thank all three reviewers for their time taken to provide valuable feedback on our manuscript, and for appreciating the quality and usefulness of our data and results presented in our study. We have improved the manuscript based on their suggestions and provide a detailed, point-by-point response below.

**Point-by-point description of the revisions**

**Reviewer #1 (Evidence, reproducibility and clarity):**
The authors have a longstanding focus and reputation on single cell sequencing technology development and application. In this current study, the authors developed a novel single-cell multi-omic assay termed "T-ChIC" so that to jointly profile the histone modifications along with the full-length transcriptome from the same single cells, analyzed the dynamic relationship between chromatin state and gene expression during zebrafish development and cell fate determination. In general, the assay works well, the data look convincing and conclusions are beneficial to the community.

Thank you for your positive feedback.

There are several single-cell methodologies all claim to co-profile chromatin modifications and gene expression from the same individual cell, such as CoTECH, Paired-tag and others. Although T-ChIC employs pA-Mnase and IVT to obtain these modalities from single cells which are different, could the author provide some direct comparisons among all these technologies to see whether T-ChIC outperforms?

In a separate technical manuscript describing the application of T-ChIC in mouse cells (Zeller, Blotenburg et al 2024, (Zeller et al., 2024)), we have provided a direct comparison of data quality between T-ChIC and other single-cell methods for chromatin-RNA co-profiling (Please refer to Fig. 1C,D and Fig. S1D, E, of the preprint). We show that compared to other methods, T-ChIC is able to better preserve the expected biological relationship between the histone modifications and gene expression in single cells.

In current study, T-ChIC profiled H3K27me3 and H3K4me1 modifications, these data look great. How about other histone modifications (eg H3K9me3 and H3K36me3) and transcription factors?

While we haven’t profiled these other modifications using T-ChIC in Zebrafish, we have previously published high quality data on these histone modifications using the sortChIC method, on which T-ChIC is based (Zeller, Yeung et al 2023)(Zeller et al., 2022). In our comparison, we find that histone modification profiles between T-ChIC and sortChIC are very similar (Fig. S1C in Zeller, Blotenburg et al 2024). Therefore the method is expected to work as well for the other histone marks.

T-ChIC can detect full length transcription from the same single cells, but in FigS3, the authors still used other published single cell transcriptomics to annotate the cell types, this seems unnecessary?

We used the published scRNA-seq dataset with a larger number of cells to homogenize our cell type labels with these datasets, but we also cross-referenced our cluster-specific marker genes with ZFIN and homogenized the cell type labels with ZFIN ontology. This way our annotation is in line with previous datasets but not biased by it. Due the relatively smaller size of our data, we didn’t expect to identify unique, rare cell types, but our full-length total RNA assay helps us identify non-coding RNAs such as miRNA previously undetected in scRNA assays, which we have now highlighted in new figure S1c .

Throughout the manuscript, the authors found some interesting dynamics between chromatin state and gene expression during embryogenesis, independent approaches should be used to validate these findings, such as IHC staining or RNA ISH?

We appreciate that the ISH staining could be useful to validate the expression pattern of genes identified in this study. But to validate the relationships between the histone marks and gene expression, we need to combine these stainings with functional genomics experiments, such as PRC2-related knockouts. Due to their complexity, such experiments are beyond the scope of this manuscript (see also reply to reviewer #3, comment #4 for details).

In Fig2 and FigS4, the authors showed H3K27me3 cis spreading during development, this looks really interesting. Is this zebrafish specific? H3K27me3 ChIP-seq or CutTag data from mouse and/or human embryos should be reanalyzed and used to compare. The authors could speculate some possible mechanisms to explain this spreading pattern?

Thanks for the suggestion. In this revision, we have reanalysed a dataset of mouse ChIP-seq of H3K27me3 during mouse embryonic development by Xiang et al (Nature Genetics 2019) and find similar evidence of spreading of H3K27me3 signal from their pre-marked promoter regions at E5.5 epiblast upon differentiation (new Figure S4i). This observation, combined with the fact that the mechanism of pre-marking of promoters by PRC1-PRC2 interaction seems to be conserved between the two species (see (Hickey et al., 2022), (Mei et al., 2021) & (Chen et al., 2021)), suggests that the dynamics of H3K27me3 pattern establishment is conserved across vertebrates. But we think a high-resolution profiling via a method like T-ChIC would be more useful to demonstrate the dynamics of signal spreading during mouse embryonic development in the future. We have discussed this further in our revised manuscript.

**Reviewer #1 (Significance):**
The authors have a longstanding focus and reputation on single cell sequencing technology development and application. In this current study, the authors developed a novel single-cell multi-omic assay termed "T-ChIC" so that to jointly profile the histone modifications along with the full-length transcriptome from the same single cells, analyzed the dynamic relationship between chromatin state and gene expression during zebrafish development and cell fate determination. In general, the assay works well, the data look convincing and conclusions are beneficial to the community.

Thank you very much for your supportive remarks.

**Reviewer #2 (Evidence, reproducibility and clarity):**
Joint analysis of multiple modalities in single cells will provide a comprehensive view of cell fate states. In this manuscript, Bhardwaj et al developed a single-cell multi-omics assay, T-ChIC, to simultaneously capture histone modifications and full-length transcriptome and applied the method on early embryos of zebrafish. The authors observed a decoupled relationship between the chromatin modifications and gene expression at early developmental stages. The correlation becomes stronger as development proceeds, as genes are silenced by the cis-spreading of the repressive marker H3k27me3. Overall, the work is well performed, and the results are meaningful and interesting to readers in the epigenomic and embryonic development fields. There are some concerns before the manuscript is considered for publication.

We thank the reviewer for appreciating the quality of our study.

Major concerns:(1) A major point of this study is to understand embryo development, especially gastrulation, with the power of scMulti-Omics assay. However, the current analysis didn't focus on deciphering the biology of gastrulation, i.e., lineage-specific pioneer factors that help to reform the chromatin landscape. The majority of the data analysis is based on the temporal dimension, but not the cell-type-specific dimension, which reduces the value of the single-cell assay.

We focussed on the lineage-specific transcription factor activity during gastrulation in Figure 4 and S8 of the manuscript and discovered several interesting regulators active at this stage. During our analysis of the temporal dimension for the rest of the manuscript, we also classified the cells by their germ layer and “latent” developmental time by taking the full advantage of the single-cell nature of our data. Additionally, we have now added the cell-type-specific H3K27me3 demethylation results for 24hpf in response to your comment below. We hope that these results, together with our openly available dataset would demonstrate the advantage of the single-cell aspect of our dataset.

(2) The cis-spreading of H3K27me3 with developmental time is interesting. Considering H3k27me3 could mark bivalent regions, especially in pluripotent cells, there must be some regions that have lost H3k27me3 signals during development. Therefore, it's confusing that the authors didn't find these regions (30% spreading, 70% stable). The authors should explain and discuss this issue.

Indeed we see that ~30% of the bins enriched in the pluripotent stage spread, while 70% do not seem to spread. In line with earlier observations(Hickey et al., 2022; Vastenhouw et al., 2010), we find that H3K27me3 is almost absent in the zygote and is still being accumulated until 24hpf and beyond. Therefore the majority of the sites in the genome still seem to be in the process of gaining H3K27me3 until 24hpf, explaining why we see mostly “spreading” and “stable” states. Considering most of these sites are at promoters and show signs of bivalency, we think that these sites are marked for activation or silencing at later stages. We have discussed this in the manuscript (“discussion”). However, in response to this and earlier comment, we went back and searched for genes that show H3K27me3 demethylation in the most mature cell types (at 24 hpf) in our data, and found a subset of genes that show K27 demethylation after acquiring them earlier. Interestingly, most of the top genes in this list are well-known as developmentally important for their corresponding cell types. We have added this new result and discussed it further in the manuscript (Fig. 2d,e, , Supplementary table 3).

Minors:(1) The authors cited two scMulti-omics studies in the introduction, but there have been lots of single-cell multi-omics studies published recently. The authors should cite and consider them.

We have cited more single-cell chromatin and multiome studies focussed on early embryogenesis in the introduction now.

(2) bT-ChIC seems to have been presented in a previous paper (ref 15). Therefore, Fig. 1a is unnecessary to show.

Figure 1a. shows a summary of our Zebrafish TChIC workflow, which contains the unique sample multiplexing and sorting strategy to reduce batch effects, which was not applied in the original TChIC workflow. We have now clarified this in “Results”.

(3) It's better to show the percentage of cell numbers (30% vs 70%) for each heatmap in Figure 2C.

We have added the numbers to the corresponding legends.

(4) Please double-check the citation of Fig. S4C, which may not relate to the conclusion of signal differences between lineages.

The citation seems to be correct (Fig. S4C supplements Fig. 2C, but shows mesodermal lineage cells) but the description of the legend was a bit misleading. We have clarified this now.

(5) Figure 4C has not been cited or mentioned in the main text. Please check.

Thanks for pointing it out. We have cited it in Results now.

**Reviewer #2 (Significance):**
Strengths:This work utilized a new single-cell multi-omics method and generated abundant epigenomics and transcriptomics datasets for cells covering multiple key developmental stages of zebrafish.Limitations:The data analysis was superficial and mainly focused on the correspondence between the two modalities. The discussion of developmental biology was limited.Advance:The zebrafish single-cell datasets are valuable. The T-ChIC method is new and interesting.The audience will be specialized and from basic research fields, such as developmental biology, epigenomics, bioinformatics, etc.I'm more specialized in the direction of single-cell epigenomics, gene regulation, 3D genomics, etc.

Thank you for your remarks.

**Reviewer #3 (Evidence, reproducibility and clarity):**
This manuscript introduces T‑ChIC, a single‑cell multi‑omics workflow that jointly profiles full‑length transcripts and histone modifications (H3K27me3 and H3K4me1) and applies it to early zebrafish embryos (4-24 hpf). The study convincingly demonstrates that chromatin-transcription coupling strengthens during gastrulation and somitogenesis, that promoter‑anchored H3K27me3 spreads in cis to enforce developmental gene silencing, and that integrating TF chromatin status with expression can predict lineage‑specific activators and repressors.Major concerns(1) Independent biological replicates are absent, so the authors should process at least one additional clutch of embryos for key stages (e.g., 6 hpf and 12 hpf) with T‑ChIC and demonstrate that the resulting data match the current dataset.

Thanks for pointing this out. We had, in fact, performed T-ChIC experiments in four rounds of biological replicates (independent clutch of embryos) and merged the data to create our resource. Although not all timepoints were profiled in each replicate, two timepoints (10 and 24hpf) are present in all four, and the celltype composition of these replicates from these 2 timepoints are very similar. We have added new plots in figure S2f and added (new) supplementary table (#1) to highlight the presence of biological replicates.

(2) The TF‑activity regression model uses an arbitrary R² {greater than or equal to} 0.6 threshold; cross‑validated R^2^ distributions, permutation‑based FDR control, and effect‑size confidence intervals are needed to justify this cut‑off.

Thank you for this suggestion. We did use 10-fold cross validation during training and obtained the R^2^> values of TF motifs from the independent test set as an unbiased estimate. However, the cutoff of R^2^ > 0.6 to select the TFs for classification was indeed arbitrary. In the revised version, we now report the FDR-adjusted p-values for these R^2^ estimates based on permutation tests, and select TFs with a cutoff of padj < 0.01. We have updated our supplementary table #4 to include the p-values for all tested TFs. However, we see that our arbitrary cutoff of 0.6 was in fact, too stringent, and we can classify many more TFs based on the FDR cutoffs. We also updated our reported numbers in Fig. 4c to reflect this. Moreover, supplementary table #4 contains the complete list of TFs used in the analysis to allow others to choose their own cutoff.

(3) Predicted TF functions lack empirical support, making it essential to test representative activators (e.g., Tbx16) and repressors (e.g., Zbtb16a) via CRISPRi or morpholino knock‑down and to measure target‑gene expression and H3K4me1 changes.

We agree that independent validation of the functions of our predicted TFs on target gene activity would be important. During this revision, we analysed recently published scRNA-seq data of Saunders et al. (2023) (Saunders et al., 2023), which includes CRISPR-mediated F0 knockouts of a couple of our predicted TFs, but the scRNAseq was performed at later stages (24hpf onward) compared to our H3K4me1 analysis (which was 4-12 hpf). Therefore, we saw off-target genes being affected in lineages where these TFs are clearly not expressed (attached Fig 1). We therefore didn’t include these results in the manuscript. In future, we aim to systematically test the TFs predicted in our study with CRISPRi or similar experiments.

(4) The study does not prove that H3K27me3 spreading causes silencing; embryos treated with an Ezh2 inhibitor or prc2 mutants should be re‑profiled by T‑ChIC to show loss of spreading along with gene re‑expression.

We appreciate the suggestion that indeed PRC2-disruption followed by T-ChIC or other forms of validation would be needed to confirm whether the H3K27me3 spreading is indeed causally linked to the silencing of the identified target genes. But performing this validation is complicated because of multiple reasons: 1) due to the EZH2 contribution from maternal RNA and the contradicting effects of various EZH2 zygotic mutations (depending on where the mutation occurs), the only properly validated PRC2-related mutant seems to be the maternal-zygotic mutant MZezh2, which requires germ cell transplantation (see Rougeot et al. 2019 (Rougeot et al., 2019)) , and San et al. 2019 (San et al., 2019) for details). The use of inhibitors have been described in other studies (den Broeder et al., 2020; Huang et al., 2021), but they do not show a validation of the H3K27me3 loss or a similar phenotype as the MZezh2 mutants, and can present unwanted side effects and toxicity at a high dose, affecting gene expression results. Moreover, in an attempt to validate, we performed our own trials with the EZH2 inhibitor (GSK123) and saw that this time window might be too short to see the effect within 24hpf (attached Fig. 2). Therefore, this validation is a more complex endeavor beyond the scope of this study. Nevertheless, our further analysis of H3K27me3 de-methylation on developmentally important genes (new Fig. 2e-f, Sup. table 3) adds more confidence that the polycomb repression plays an important role, and provides enough ground for future follow up studies.

Minor concerns(1) Repressive chromatin coverage is limited, so profiling an additional silencing mark such as H3K9me3 or DNA methylation would clarify cooperation with H3K27me3 during development.

We agree that H3K27me3 alone would not be sufficient to fully understand the repressive chromatin state. Extension to other chromatin marks and DNA methylation would be the focus of our follow up works.

(2) Computational transparency is incomplete; a supplementary table listing all trimming, mapping, and peak‑calling parameters (cutadapt, STAR/hisat2, MACS2, histoneHMM, etc.) should be provided.

As mentioned in the manuscript, we provide an open-source pre-processing pipeline “scChICflow” to perform all these steps (github.com/bhardwaj-lab/scChICflow). We have now also provided the configuration files on our zenodo repository (see below), which can simply be plugged into this pipeline together with the fastq files from GEO to obtain the processed dataset that we describe in the manuscript. Additionally, we have also clarified the peak calling and post-processing steps in the manuscript now.

(3) Data‑ and code‑availability statements lack detail; the exact GEO accession release date, loom‑file contents, and a DOI‑tagged Zenodo archive of analysis scripts should be added.

We have now publicly released the .h5ad files with raw counts, normalized counts, and complete gene and cell-level metadata, along with signal tracks (bigwigs) and peaks on GEO. Additionally, we now also released the source datasets and notebooks (Rmarkdown format) on Zenodo that can be used to replicate the figures in the manuscript, and updated our statements on “Data and code availability”.

(4) Minor editorial issues remain, such as replacing "critical" with "crucial" in the Abstract, adding software version numbers to figure legends, and correcting the SAMtools reference.

Thank you for spotting them. We have fixed these issues.

**Reviewer #3 (Significance):**
The method is technically innovative and the biological insights are valuable; however, several issues-mainly concerning experimental design, statistical rigor, and functional validation-must be addressed to solidify the conclusions.

Thank you for your comments. We hope to have addressed your concerns in this revised version of our manuscript.

**Author response image 1. sa3fig1:** (1) (top) expression of tbx16, which was one of the common TFs detected in our study and also targeted by Saunders et al by CRISPR. tbx16 expression is restricted to presomitic mesoderm lineage by 12hpf, and is mostly absent from 24hpf cell types. (bottom) shows DE genes detected in different cellular neighborhoods (circled) in tbx16 crispants from 24hpf subset of cells in Saunders et al. None of these DE genes were detected as “direct targets” in our analysis and therefore seem to be downstream effects. (2) Effect of 3 different concentrations of EZH2 inhibitor (GSK123) on global H3K27me3 quantified by flow cytometry using fluorescent coupled antibody (same as we used in T-ChIC) in two replicates. The cells were incubated between 3 and 10 hpf and collected afterwards for this analysis. We observed a small shift in H3K27me3 signal, but it was inconsistent between replicates.

References

Chen, Z., Djekidel, M. N., & Zhang, Y. (2021). Distinct dynamics and functions of H2AK119ub1 and H3K27me3 in mouse preimplantation embryos. Nature Genetics, 53(4), 551–563. den Broeder, M. J., Ballangby, J., Kamminga, L. M., Aleström, P., Legler, J., Lindeman, L. C., & Kamstra, J. H. (2020). Inhibition of methyltransferase activity of enhancer of zeste 2 leads to enhanced lipid accumulation and altered chromatin status in zebrafish. Epigenetics & Chromatin, 13(1), 5.

Hickey, G. J., Wike, C. L., Nie, X., Guo, Y., Tan, M., Murphy, P. J., & Cairns, B. R. (2022). Establishment of developmental gene silencing by ordered polycomb complex recruitment in early zebrafish embryos. eLife, 11, e67738.

Huang, Y., Yu, S.-H., Zhen, W.-X., Cheng, T., Wang, D., Lin, J.-B., Wu, Y.-H., Wang, Y.-F., Chen, Y., Shu, L.-P., Wang, Y., Sun, X.-J., Zhou, Y., Yang, F., Hsu, C.-H., & Xu, P.-F. (2021). Tanshinone I, a new EZH2 inhibitor restricts normal and malignant hematopoiesis through upregulation of MMP9 and ABCG2. Theranostics, 11(14), 6891–6904.

Mei, H., Kozuka, C., Hayashi, R., Kumon, M., Koseki, H., & Inoue, A. (2021). H2AK119ub1 guides maternal inheritance and zygotic deposition of H3K27me3 in mouse embryos. Nature Genetics, 53(4), 539–550.

Rougeot, J., Chrispijn, N. D., Aben, M., Elurbe, D. M., Andralojc, K. M., Murphy, P. J., Jansen, P. W. T. C., Vermeulen, M., Cairns, B. R., & Kamminga, L. M. (2019). Maintenance of spatial gene expression by Polycomb-mediated repression after formation of a vertebrate body plan. Development (Cambridge, England), 146(19), dev178590.

San, B., Rougeot, J., Voeltzke, K., van Vegchel, G., Aben, M., Andralojc, K. M., Flik, G., & Kamminga, L. M. (2019). The ezh2(sa1199) mutant zebrafish display no distinct phenotype. PloS One, 14(1), e0210217.

Saunders, L. M., Srivatsan, S. R., Duran, M., Dorrity, M. W., Ewing, B., Linbo, T. H., Shendure, J., Raible, D. W., Moens, C. B., Kimelman, D., & Trapnell, C. (2023). Embryo-scale reverse genetics at single-cell resolution. Nature, 623(7988), 782–791.

Vastenhouw, N. L., Zhang, Y., Woods, I. G., Imam, F., Regev, A., Liu, X. S., Rinn, J., & Schier, A. F. (2010). Chromatin signature of embryonic pluripotency is established during genome activation. Nature, 464(7290), 922–926.

Zeller, P., Blotenburg, M., Bhardwaj, V., de Barbanson, B. A., Salmén, F., & van Oudenaarden, A. (2024). T-ChIC: multi-omic detection of histone modifications and full-length transcriptomes in the same single cell. In bioRxiv (p. 2024.05.09.593364). https://doi.org/10.1101/2024.05.09.593364

Zeller, P., Yeung, J., Viñas Gaza, H., de Barbanson, B. A., Bhardwaj, V., Florescu, M., van der Linden, R., & van Oudenaarden, A. (2022). Single-cell sortChIC identifies hierarchical chromatin dynamics during hematopoiesis. Nature Genetics. https://doi.org/10.1038/s41588-022-01260-3